# Seed Systems Resilience—An Overview

**Niels P. Louwaars** [1,2,]*  and **Gigi Manicad** [3]

1   Plantum, Vossenburgkade 68, 2805 PC Gouda, The Netherlands
2   Law Group, Wageningen University, P.O. Box 102, 6700 AC Wageningen, The Netherlands
3   Independent Expert, 2543 BL The Hague, The Netherlands
*   Correspondence: n.louwaars@plantum.nl

**Abstract:** Seeds are a basic input for all crop production. Good seed is crucial for the global food and nutrition security, for rural development and farmers' livelihoods and for all green value chains. What "good seed" is depends on individual farmer's needs. The sustainable availability of seed and seed choice is thus an essential issue for every farmer. Farmers access seeds from multiple sources. Different seed systems have their strengths and vulnerabilities. With changing farming conditions around the world due to climate change, soil degradation and market demands, an analysis of possible disruptions and general future-proofing appears necessary. Such analysis also informs the kinds of support that different seed systems may need to be optimally resilient. Given the very limited studies on resilience of current seed systems, we performed a literature review on the opportunities and vulnerabilities of seed systems to identify possible resilience challenges. Seed system resilience in terms of future-proofing is closely linked with "seed security", the secured access to preferred seeds by farmers. Such resilience depends on the functioning of each of the formal and farmers' seed systems and how these two systems complement each other, particularly when one falters. In this paper, we describe the major characteristics of seed systems, analyze their strength and vulnerabilities, and describe support functions toward future-proofing them. Both the farmers' and the formal seed systems are increasingly facing varying forms and degrees of sustainability challenges. These require various types of support. Farmers' seed systems may find it hard to respond to radically changing conditions without support. Commercial formal seed systems are less vulnerable, as they generally are better equipped to respond to the (changing) needs of their customers. Commercial formal seed systems, however, only serve those farmers that are able and willing to pay for quality seeds. This immediately indicates a significant limitation. A common feature of both formal and farmers' systems is their dependence on the continuous flow and capacity to use a diversity of plant genetic resources. In conclusion, no single seed system can be considered totally resilient and thus future-proof without specific external support.

**Keywords:** formal seed systems; farmers' seed systems; resilience; seed security; commercial seed; seed cooperative; see policy; seed regulations

## 1. Introduction

All farmers need good seeds for every planting season. Seeds are a basic input for all crop production, as such seeds are basic to food security. Seeds are also fundamental for the livelihood strategies of farming communities worldwide. The word "seed" is used in this paper as to include both true botanical seeds that form after sexual recombination, and vegetative planting materials, stem cuttings (e.g., cassava) and potato tubers. We focus on the seed systems of domesticated food crops, mainly staples and vegetables but excluding (fruit) trees. The term "seed system" is defined as the combined activities of actors, making use of plant materials and knowledge, that together are necessary for suppling seeds to farmers [1]. A functioning seed system should sustainably enable farmers to have access to the seeds of their choice and purpose, of the best possible qualities, the right time and at the right price from the farmers' investment perspectives. Farmers need access to seeds at

the start of each cropping season, and with a longer-term perspective. All systems include aspects of breeding/selection, production/multiplication and distribution in more or less complex settings. These include the corresponding distribution of tasks between multiple operators, including seed policy development and implementation [2].

The terms formal and informal (or farmers') seed systems are commonly used to distinguish between seed systems. Farmers' seed systems are managed in all their components by farmers [3], whereas formal systems operate through professionals for each component, various aspects of which are commonly regulated [4]. The seed production component is, however, performed by farmers, either on their own account or under contract with the seed enterprise. There are significant differences within both types of seed systems, and intermediate ones exist where some components are more formal, and others operated by farmers themselves. Various components of the formal sector are operated either by public or private entities. For example, private seed production and distribution may be conducted using varieties bred by public research organizations; quality control may be performed by a combination of public and private operators [5].

Any farmer may use at any time different seed systems [6,7]. Vegetable seeds for commercial production may be accessed from an internationally operating company, maize from commercial national seed production, sorghum seed from the local brewery, sesame seeds from the produce exporter, cowpea seeds from the local food market, cassava cuttings from a neighbor, and millet seeds saved on-farm. For all seeds, a farmers will decide their source on the basis of availability, perceived quality aspects and cost. Resilience of each system is essential for farmers, as, in the example above, the commercial seed supplier may not offer cowpea seeds and the saved sesame may not comply with the quality requirements of the market.

## 2. Methods

The importance of seed systems has received significant attention in the literature lately, including finding complementarities among the formal and the so-called informal or farmers' seed sectors. This has particularly led to promoting integrated seed sector development. Much of the focus has been on addressing present constraints in order to increase farmers' access to superior varieties. However, with fast and vast climatic and market changes, there is a gap in the analysis and interventions on the vital elements for future-proofing both seeds systems, individually and together.

This paper is based on a literature review and the largely complementarity experiences of the two authors in different seeds systems in a wide range of countries in Asia, Latin America, Africa and Europe. The major source was Wageningen University library, which provides access to the major databases for agricultural and plant sciences (including seeds and seed systems), social science (and specific participatory and development economics) and environmental (resilience) literature. This search was complemented with unpublished sources from the authors' own network in seed sector development, including surveys, evaluation and seed programs that directly involved farmers. We necessarily had to streamline the massive literature data to contribute to the focus on this review: resilience of seed systems.

The term resilience is used here as the capacity of the seed system, as a whole, to anticipate, absorb, respond or transform due to changes. Such changes can come in the form of threats or opportunities brought about by markets, the environment and technologies. A resilient seed system contributes to the sustainability of seed supply and demand within a given context, and thus to seed security. Seed security can also be sustained when the formal and farmers' seed systems can complement each other, particularly when one of them falters, hence increasing seed resilience through greater diversity of crops, varieties and seed sources. For example, this can happen when seeds from the formal systems are added to or replace farmers' seeds within their systems of seed production and exchange. Or the other way round, when farmers' seed systems fill gaps that are left by the formal seed supply.

We analyzed the seed system literature using a resilience framework, even though most of the articles were not written with that focus. The three key elements for the functioning of the seed systems are: (1) continuous access to diversity in crops and varieties; (2) institutions pertaining to norms, regulations, systems of exchange and social relations; and (3) innovation related to plant breeding.

After a brief introduction of the major seed systems, we analyzes their strengths and main vulnerabilities, which may have temporary or lasting effects on their resilience in changing conditions. We framed the analysis based on the three key elements of seeds systems resilience and described support functions toward future-proofing them.

## 3. Seed Systems

### 3.1. Farmers' Seed Systems

Farmers' seed systems, also often referred to as informal, or farmer-managed seed systems, include the selection and production of seeds by farmers themselves, commonly as part of their normal crop production operations, storage onto the next planting season, and the sharing or local sales of seeds. Over the millennia, farmers have been selecting, domesticating, developing and conserving seeds, which have formed the basis of our global food system. To understand farmers' seed systems, it is important to understand the shifting context and livelihood strategies, within which these particular seed systems evolve and function. Farmers' seed systems cater to highly diverse peoples, within highly diverse agro-ecologies and diverse cropping systems. Such farmers often have small land holdings with varied tenure rights, within marginal or high potential environments. Land holdings may be parceled into two to three locations, sometimes with different soil types and micro-climates, which may require different varieties. Within a family, there are divisions of tasks between men and women, including in those related to seed. Men and women may have different access to land, seeds, market and labor engagements. As a result, men and women may have different crop and varietal preferences. Farmers may thus manage a portfolio of crops and crop varieties that may include domesticated, semi- and non-domesticated ones [8]. Farmers seed systems are characterized by a diversity of traditional and/or formal sector varieties that are highly adapted within specific ago-ecologies. These plants have been developed though processes of human and natural selection in specific environments, farming practices, and farmers' preferences. Farmers' selection is based on farmers' knowledge of the (diversity within a) variety and quality perceptions for cultivation and culturally important traits [9]. Either by necessity, or because many farmers like "things to try", they may access seeds from elsewhere [3]. Farmers are also generally willing to share their seeds with others when asked for a sample; some may only share seeds with people they know. This sharing and testing of the new variety may result in the replacement of the farmers' own varieties, or it may become part of it, through admixture or natural crosses. This is also how scientifically bred varieties from the formal sector may find their way into the farmers' seed system. As seeds are an "experience good", the farmers' seed systems are thus part of the social fabric of communities and relationships between them and thus highly context specific [10].

Farmers' seeds systems largely cater to the complexities of about five hundred million small-holder farmers worldwide. FAO/IFAD/WFP [11] claim that 80% of seeds that small holder farmers access are from the farmers' seeds systems, half of which are sourced from local grain markets (see Section 3.3), and 55% are paid in cash. This indicates that many farmers are willing and able to pay for seeds within specific conditions, which still needs to be understood. The 25% remaining are farm-saved seeds or obtained through farmer-to-farmer sharing.

Poorly performing seeds can be devastating for the livelihood of farmers since seed is an "experience good", i.e., their performance cannot always be assessed merely by looking at it, but only once they have been planted and grown. "Trust" is, therefore, a vital element in the functioning and resilience of any seed system. Institutionally, trust in

seed quality and reliability of the source can be endowed through social relations and/or regulatory systems.

*3.2. Formal Seed Systems*

Seed systems involving different professionals started to develop in Europe in the 18th century [12]. Notably, the production of vegetable seeds, i.e., crops that are not produced for their seeds such as cereals and pulses, has created a particular challenge for farmers. Seed production is then separated from crop production. Some plants have to be kept in the field in order to flower and produce seeds, which may disturb land preparation for the next crop. Similarly, livestock farmers are not commonly geared to producing forage seeds, creating a demand for seed produced by others. Seed markets developed, creating a further specialization of tasks as a result of technical developments in selection and seed technology. This creates a demand for seed, a "pull" for specialist producers of good quality seed of specific varieties. Both the International Seed Testing Association and the International Seed Federation were established in 1924 to support international seed trade [13]; the first aimed at standardizing seed testing; the second was an initiative of mainly forage seed traders, but with a widening focus to cover all seeds.

Further opportunities arose with the emergence of plant sciences. These enabled the creation of diversity and the selection within that diversity, which are the basic components of plant breeding, notably in Europe in the late 17th century [12]. The clarification of heredity by Mendel, which became known to the scientific community after the rediscovery of Mendel's writings in 1900, created a tremendous "push" toward plant breeding and the formal seed markets. A similar "push" created an interest in seed production during the Green Revolution from the 1960s onward. Large scale investments in many countries supported by agencies such as the World Bank and FAO thus created ways to obtain new plant varieties multiplied and accessed by farmers [14].

Both these "push" and "pull" aspects in seed chains continue to exist today with specialized breeders, seed producers, seed conditioners and marketers, jointly forming the formal seed system. The term "formal" particularly refers to the need to create rules for the marketing of seed following increasing distances between the producers and the users of seeds. The quality aspects of most seed, as well as the identity of the variety that defines expectations for yield and product quality, cannot be observed by merely looking at the seed. Therefore, a reliable label that duly informs the buyer has become increasingly essential for seeds as an "experience good". The reliable label is based on seed testing and varietal identity preservation in the chain from breeder to seed market. The first official seed laboratory was established in Germany 1869 by Friedrich Nobbe [15]. Labels on the seed bags, based on standardized procedures and quality standards, replaced the "trust" put into the seed seller in local seed sharing. Naming of varieties based on a detailed description became standard requirements for many crop seeds. Such standards required a degree of homogeneity in important characteristics, which also allowed for the protection of breeder's rights. This also means that varieties had to be stable in the expression of their main traits over several generations of multiplication. Furthermore, farmers' interest in knowing more about new varieties led to variety testing for their value for cultivation and use (VCU) and demonstration fields. Many of these functions also became regulated in order to protect and inform farmers.

There are significant differences in the distribution of tasks between the public and private sectors in formal seed chains in different geographies and crops. In Europe, plant breeding originated in the private sector and still is almost completely private even though governments invest significantly in breeding research. In the United States, on the other hand, breeding of most crops with the exception of the most commercial seed products is still performed in the public sector at the Land-Grant Universities [16]. Plant breeding can be performed in the private sector for those crops for which a profit margin on the sale of seeds warrants investments in research. This is also the reason why in the Global South,

most breeding is performed by national public research institutions, for the major crops with support from the Consultative Group on International Agricultural Research (CGIAR).

In most of the Global South, formal seed production started off as a public operation. There are some exceptions, such as in Kenya and Zimbabwe, where cooperatives of large farmers invested in seed production and breeding. For most other countries, state farms and government-led contract grower schemes were set up, and seed conditioning plants were erected with government or donor funding. Seed distribution was handled by the government extension schemes in many countries. Most of these seed schemes, set up with donor funds [17,18], have been privatized following structural adjustment policies in the 1990s [19,20]. Some countries invited international companies to take over the facilities (e.g., Malawi), others sold them to local investors (e.g., Uganda), and yet others put the government seed operations at some distance from public governance (e.g., Ethiopia). Despite the private nature of these companies, some "protection" by the governments remained because seed is considered a strategic good for the country.

During the process of privatization of seed production, the issue of quality assurance comes up. In many government-run seed operations, variety registration, seed testing and certification were an integrated part. However, in a privatized sector, seed quality control organizations had been developed to make independent assessments. These commonly operate within the government, but differences among countries exist with regard to compulsory and voluntary rules [16] and between implementation within the government organization, or by private labs performing such public tasks. It is particularly this independence of quality control and seed producers that requires a formal backing through seed regulations that shape the formal sectors [4].

### 3.3. Other Seed Sources

The distribution of free seed, notably by non-governmental organizations, but also by governments and intergovernmental agencies, is becoming quite significant. This is performed either as a response to crises, such as natural or human-made disasters [21–23], or in the frame of supporting farmers through free or heavily subsidized inputs [24].

Finally, many near-subsistence farmers do not save their own seed or obtain it in the community but rely on purchasing food grain from the local market to plant their next crop. Statistics are rare and incomplete, but FAO studies indicate that up to 20–40% of farmers in West Africa use food grain as seed [25]. They are too poor or otherwise unable to select and store seeds themselves.

## 4. Strengths and Vulnerabilities

### 4.1. Farmers' Seed Systems

Farmers' seed systems are dynamic; they are based on farmers' intimate knowledge of their production conditions, their varieties and their responses to these conditions. They furthermore value the product qualities of their crops for consumption and for cultural uses. Since they have harvested the seeds themselves, or at least saw the crop when they source the seeds within the community, they know by experience what to expect in terms of seed quality. This depends, for example, on the weather during harvesting. In addition, the crop is likely to perform similarly compared to the previous one, since the variety can be considered adapted to the local conditions [7]. In the case of genetically diverse varieties, it may be expected that this diversity provides a level of resilience of the variety toward different and changing production conditions. Scientifically bred uniform varieties also commonly find their way to farmers through informal exchange systems and are multiplied in the same way. They may be cultivated in a pure stand or mixed with other seeds [10].

Farmers' seeds systems have been evolving through time with changing environments and market changes. A 2013 baseline study in Vietnam, Peru and Zimbabwe showed that farmers are aware of climate change based on impact to their farms and crop performance. To adapt, they shifted to crops and crops varieties with greater resilience and changed

planting times. They also combined crops and varieties with varying stages of maturity as a risk management measure. These strategies were effective but up to a point [26].

In the Philippines, indigenous peoples in the Ifugao Mountain province plant both indigenous varieties of rice primarily for superior and highly preferred eating quality and commercial rice varieties that are not photo-sensitive, enabling them to have two seasons and greater harvest for selling in the markets. They also grow hybrid vegetables from commercial companies as cash crop [27]. Farmers' networks and distribution channels proved quick to recover from shocks in Rwanda [28].

Vulnerabilities

Given the seasonality of agriculture and that most small-holder farmers operate under rain-fed conditions, they are particularly vulnerable to environmental hazards such as droughts, flood, pest and diseases, and erratic weather patterns. They are vulnerable to not only losing their crops but also to losing their seeds. Second, recurrent and predictable food shortages occur during lean or hunger periods. Very vulnerable households may resort to eating pre-mature crops and their seeds for the next growing season. For example, overharvesting of perennial crops such as taro is likely to result in reduced yield, or the consumption of cereal seeds is a major threat to the seed security of farmers [29]. Third, vulnerabilities are magnified when environmental calamities occur during the hunger period and during recurring environmental calamities such as the recuring 20-year drought in Zimbabwe, followed by flooding. These often do not allow farmers to recover from one disaster to the next and could seriously erode their assets such as seeds and their social networks. Women, despite their role in seed management, are particularly more vulnerable given their marginalized positions. Women tend to have the double burden of taking care of the farm and households, and at the same time, they generally have less decision making and less access to resources.

—A basic given of local seed supply is that it is anti-cyclic. After a good season, farmers have enough good seed, and thus have their neighbors. After a bad cropping season or very poor harvesting conditions, however, yields and seed qualities may be low, and surpluses to save as seed will be scarce. A farmer who might normally rely on the community in case of shortages may have a problem, because the neighbors are likely to face similar challenges. Seed security can thus be an issue [30]. That is aggravated by climate-induced, recurring and/or more severe natural disasters. The same is true for protracted human-made disasters and (temporary) displacements, which may affect both formal and farmers' systems. However, farmers' seed systems are likely to be severely more impacted.

—The occurrence of plant diseases and pests, several of which are seed transmitted. Farmers have good knowledge about many aspects of their seed. Identification of seed-transmitted diseases is, however, limited. Little is known in farming communities about the origin of spots on the leaves or other symptoms, let alone which spots may be seed transmitted. When the disease shows on the harvested seed, such as Anthracnose in beans, hand sorting can remove most of the diseased seeds. Otherwise, this provides the disease agents a quick start to develop epidemic proportions in the following seasons. This is a major limitation in on-farm potato multiplication and happens in many other crops. In other crops, an accumulation of plant diseases is common in farmers' seed systems.

—Without continuous selection, farmers' varieties can degenerate, weakening positive traits, productivity and quality. Preferred qualities such as aroma, texture and taste, as well as pest and disease resistance, can be lost over time due to a mix of factors such as introgression, mutation, admixture.

—Whereas the genetic diversity of crops provides a certain resilience in different weather conditions and disease occurrences across growing seasons, such genetic dynamics in the crop may not always be able to cope with quick ecological changes; evolution is notoriously slow; thus, changing crops may be the quickest remedy. The occurrence of severe droughts and erratic rainy seasons put the resilience of farmers' varieties to the test

in current climate change situations. Even more dangerous is the spread of "new" diseases that may evolve naturally or enter a "new" region as another effect of climate change or international trade. The outbreak of UG 99 in wheat is an example of such a disease for which the local genetic diversity was not prepared for, and new resistant varieties had to be developed [31].

Breeding as part of the farmers' cropping systems is severely limited by a lack of knowledge and opportunities to obtain new diversity, or facilities to develop complex crossing programs and variety testing. Farmer breeding can include crosses, notably by planting unrelated plants of cross-fertilizing crops such as maize next to each other. Beyond that, most variety development work is based on (mass) selection and occasional introductions, of other local or scientifically bred materials. Sharing of materials is, in many cases, limited to relatives and neighbors—sharing across natural (valleys) and cultural (tribe) barriers may be limited [32].

In summary: farmers' seed systems have proven to be important for very many farmers during changing conditions. Challenges in terms of resilience relate to vulnerabilities to an accumulation of shocks (disasters and a more general anti-cyclic nature) and more gradual specific ecological and market changes due to climate change and urbanization. This relates to demand and availability of seed, but particularly to the adaptation of the farmers' varieties and access to new ones.

### 4.2. Formal Seed Systems

Effective formal seed systems should be able to offer good quality seeds of locally adapted varieties for every planting season. Scientific breeding should be able to provide solutions for the major agro-ecological challenges of the financially viable farmers and provide a good and uniform product that meets the expectations of the market and qualities for home consumption. The farmers who obtain seeds through the formal channels can be confident of the quality assurance by an identity mark on the label. Formal systems, and notably those that span large geographies, are less challenged by local weather problems—less than expected seed yields in one area can be compensated for by good yields obtained elsewhere. Similarly, breeding in international organizations can benefit from scale advantages. New technologies and pre-breeding operations can be centralized, while the final steps of breeding and the testing of varieties need to be performed under the diverse local conditions. Furthermore, controlled storage, regular testing, and innovations in seed processing and treatments can be applied in the formal sector, securing seed quality.

Vulnerabilities

—A basic challenge is the economic sustainability of the system. The added cost of producing quality seed compared to food grain poses limits for the commercial seed producers.

Seed production requires more care than crop production, which adds cost. Centralized storage, treatments, transport and packaging, and locking up vast amounts of money between the production of seeds and sales are all cost factors. Additional costs are incurred in marketing/distribution, seed quality management, and overheads. A rule of thumb is that formally produced seed (of self-fertilizing crops) needs to be at least 1.5–2 times as expensive as food grain of the same crop to break even. This calculation does not take into account the heavy investments in research and breeding. These accumulated costs mean relatively high prices for the farmers. This may not be a problem when significant financial benefits can be obtained compared to other options, notably farm-saved seed. This is easier to obtain with crops that are not commonly grown for their seed such as vegetables and forages, and for hybrid crops where the benefit of the hybrid vigor outweighs the cheaper price of open-pollinated alternatives. Such seed production can be profitable, but this commonly translates into higher investments in R&D to keep ahead in such competitive markets.

—Government seed production schemes, which have been developed in the frame of the Green Revolution, have proven to be unsustainable.

Government seed production schemes were developed in many countries to take the products of breeding to as many farmers as possible. However, the government objective to keep seed prices affordable for farmers has not been economically sustainable for large-scale public seed production, which has disappeared almost everywhere. Private seed production does not have this limitation, but only if their market can afford the seed prices that can make them at least break even. Public seed production schemes can even harm the emergence of a competitive viable private sector, and so do seed subsidies when not efficiently targeted. Seed producers for important food crops, however, remain dependent on public breeding and a regular supply of (pre-)basic seeds from the breeding institutes, which is an important contribution to increasing farmers' choice. However, such institutes commonly find it hard to give enough attention to variety maintenance [33,34]. Such work creates neither scientific publications nor new varieties, criteria that breeders are commonly rewarded for, which would warrant additional support.

—Breeding requires a long-term investment and a clear vison of future needs.

Breeding is a very powerful tool to adapt crops to the needs of farmers and consumers. It is also an essential prerequisite to adapt crops to any change in cropping system and market requirement. A general challenge for breeders is that they have to concentrate on a sufficiently large "recommendation domain" [35], even though needs of different farmers may be quite diverse [36]. This is true for both public and private breeders: both public and private investments have to be used in the most efficient ways.

Breeding thus also has to anticipate future developments and needs, commonly in a time frame of 10–20 years, which is by default a basic contribution to system resilience. This requires intimate contact with the (prospective) customers and a vison of future needs based on trends in both agro-ecologies and markets. This is a particular challenge in public sector breeding, which remains essential for crops and farmers that the private sector cannot invest in. Public policies focus both on food security and rural development. The former is met by breeding major food crops (which are often not profitable seed crops) and have a focus on efficient commercial farmers; the second objective relates to the many resource-poor farmers. Assessing true needs of smallholders is further complicated since public extension services have now significantly declined in many countries. Research managers in the public sector may also find it difficult to assess the performance of their breeders. Should this be based on publications like the other scientists, on the number of varieties that are formally released, on the number of released varieties or on actual acreage planted to the breeder's varieties, or more specifically on the success of their varieties in farmer-to-farmer seed exchange? Private breeding teams can be assessed on the basis of their current and prospective market share. Judging public breeders based on licenses income on varieties, which is performed in some places, may go against public policies to cater to the needs of the majority of farmers.

—Breeding is an interdisciplinary process.

On the one hand, breeders are depicted as providing solutions to major global challenges; at the same time, breeders know that they cannot meet such expectations on their own. Even where breeding programs may be well catered for financially, e.g., through breeder's rights or the sales of breeders' seed, breeders know that they need others. Breeders need social scientists to help prioritize breeding goals, as well as pathologists, agronomists, and biotechnologists to develop new varieties. This needs to be taken into account by those deciding on funding.

—Official quality controls are difficult to sustain.

Large scale seed production requires quality control throughout the seed-production processes. International linkages in breeding and seeds require effective phytosanitary services. The varietal identity has to be maintained, and seed qualities such as germination percentage and seed health need checking. Farmers' seed supply is generally built on trust, but formal seed systems operate at a distance from the customer, who have to rely on a label.

The quality management required for a reliable label involves independent inspectors and laboratories with significant logistics, facilities and human resources. Poorly functioning systems result in sub-standard seeds entering the market. This may even invite fake seeds, grain in nice bags sometimes with a lemonade coloring to mimic treated seed, creating significant damage to farmers. The most sustainable way is to charge the seed producer the cost of such operations, but this challenges the economic viability of seed production of major food crops and thus defies the objective. Introducing a quality declared seed class can reduce costs especially for remote seed producers with a limited geographical focus, such as farmer–seed cooperatives.

*4.3. Other Seed Sources*

Obtaining "grain" from the local food market for use as seed may be quite sustainable from an availability and price point of view, but it carries big risks of poorly adapted varieties, low germination and vigor and presence of seed-transmitted diseases. When the grain originates from the same locality, the varietal characteristics may still be fine, and type names may even be known by the merchants. Smell and color can be indicative, but the poorest farmers run great risks of using food grain as seeds.

The distribution of free seed to support farmers to boost production may be well meant. However, this may make farmers dependent and may obliterate their own seed selection and production capabilities, which is particularly important in resource-poor farming systems. Subsidizing seeds can only be sustainable when the donor or government is able and interested to continue the funding for a very long time. Free distribution of small seed samples for farmers to try out a new variety is however a different issue, which may be a useful way to introduce farmers to new options.

## 5. Options for Increasing Farmers' Seed Systems Resilience

Challenges toward farmers' seed systems' resilience relate to vulnerabilities to multiple changes and recurring shocks (disasters and general anti-cyclic nature), which restrict the time for recovery, and which further erode farmers' assets such as seed materials and social networks. These challenge seed availability and more gradual specific ecological and market changes due to climate change and urbanization, challenging particularly varietal adaptation. Making such systems more resilient includes various activities that farmers can perform themselves and that require cooperation with the formal sectors.

—Strengthen the system that enables farmers to continue to adapt and innovate within their changing context.

Given the contextual diversities and complexities of the farmers' seeds systems, a simplistic and fragmented approach of prescribing seeds replacement [37] should be cautioned with the "do no harm" principle. Despite serious challenges, farmers' seed systems have proven crucial to the continuing innovation and functioning of their seed supply. On the other hand, interventions that restrict farmers to only traditional crops and varieties may be ignoring farmer's shifting practices and preferences and the changing environments markets. In this regard, it would be more effective and participatory to support both the farmers and their systems to continue to innovate and adapt to their changing circumstances and preferences. This would involve strengthening farmers' capacities for their continuing access and use of a diversity of seeds and planting materials. This would also involve strengthening social networks for the local sharing of seeds and knowledge, and their improved organization to jointly articulate and demand for seed materials and support services and participate in related policy processes and markets. To support the resilience of farmers seeds systems, below are more specific recommendations.

—Facilitate participatory diagnostics and support farmers' decision making.

Farmers within communities and within households may have different sets of complex and shifting criteria for their seed selection, enabling farmers to choose which (combination of) seeds usually need to be facilitated by participatory diagnostics and decision support tools [38]. Most effective are the methods that draw on farmers' knowledge and

engage them in experiential learning such as with farmer field schools where farmers learn from each other [39]. However, many participatory tools still need to be calibrated for gender sensitivity, and special focus groups for women may be necessary, notably in dealing with seeds, where in many situations, women have particular roles in seed selection, storage, exchange and in breeding.

—Ensure the farmers' continuing access and sustainable use of a diversity of seeds.

The need to access additional and novel plant diversity in farmers' seed systems is also broadly recognized [26,40]. This plant diversity may come from both farmers' landraces and from the formal sector; these may be segregating or stable lines. Farmers' continuing access to diversity is closely linked to farmers' capacities to make informed choices among crops, varieties and breeding lines.

Initiatives to improve local varieties rather than replacing them started in the early 1990s. Such "participatory plant breeding" initiatives were initiated both from breeders [41,42] and from a more social scientist [43] perspective, specifically linking it to seed systems [44]. There are also cases that the genetic resources objectives are the major driver of interventions [45]. They can bring in crop traits from outside to enrich the farmers' varieties and select them based on the farmers' knowledge in their locality. A major challenge of this concept relates to limitations for upscaling and outscaling, which limit the efficiency of the science input [46,47]. However, a FAO review [48] reports on promising models where multi-stakeholders and multi-country collaboration and capacity building facilitated the farmers access to about 20,706 varieties, which were characterized and/or tested for the development and adaptation in multiple locations around the world. This in turn generated the selection and development of 298 new varieties. In addition, 5933 new accessions can be accessed for further development and adaptation for more climate-resilient plant genetic resources for both formal and farmer seed systems.

—Strengthen farmers' capacities for seed selection and storage.

Farmers' capacities to select seeds from standing crops vary greatly. Improving capacities for seed section tend to be easy with immediate and tangible effects for the farmers' technical capacities and confidence. Participatory variety selection programs can, in line with participatory plant breeding, increase both the diversity that farmers can select from and their selection capabilities [49,50]. For seed storage, ensuring that seeds are well sorted so as to avoid mixing varieties, as well as ensuring that seeds are properly dried, are important steps. Furthermore, good seed storage facilities at the farm level are a basic prerequisite. Improved seed storage structures protect against rodents, and airtight bags provide good protection against insects and can reduce life processes in the seed as soon as the oxygen in the bag is depleted [51].

Diversify farmers' seed storage facilities and strengthen linkages within and among communities and with institutions from the formal system.

Community seed banks can be important to improve resilience [52]. Seed banks may compliment, not replace, individual household seed storage, especially in times of stresses [53]. For community seed banks to work sustainably, they should be an integral part of the farmers' seed management, whereby the usefulness of the community seed bank outweighs the operations and maintenance cost. Technically, the community seed banks should provide services for the portfolio of seeds that farmers choose as important and need to be in-stored. The services include seed storage and multiplication, ensure minimal seed quality though testing and regeneration, and have a catalogue of seed collection and characterization. Community seed banks require a strong commitment and community governance. A network of such banks and/or community bio registers, combined with seed fairs [54,55], can also exchange seeds or retrieve lost seeds with other communities from relatively medium to far distances, hence potentially helping to resolve the anti-cyclic nature of seed supply. Examples exist with a long history of successful operation [52], but there are also largely undocumented failures. Community seedbanks are often combined with community genebanks that can conserve and make small quantities of local genetic resources available for multiplication in case of need [56]. They can also link up with

government agencies, such as extension services and national and international genebanks, which could keep sample duplicates for safe keeping, repatriate such samples when needed and introduce new samples [48].

—Support farmers to diagnose and manage seed-transmitted plant pests and diseases. Plant clinics, operated by extension agents with back up from specialist laboratories, are excellent for diagnosis, surveillance and disease management in smallholder farmer areas. The concept has been implemented in a wide range of countries [57,58]. It is yet too early to assess their sustainability. Early identification of seed-transmitted diseases in such clinics can provide farmers with important information to which plants can produce seeds and to which food.

## 6. Options for Increasing the Resilience of the Formal Seed Systems

—A basic challenge is economic sustainability of seed production. The added cost of producing quality seed compared to food grain limits commercial seed producers in the crop seeds that they produce.

The production of quality seed requires care such as centralized storage, treatments, transport, and packaging, which lock up vast amounts of financial resources. Administration, marketing, seed quality checks, and management add to the cost of seeds [59]. Large companies have some scale advantages, but commonly also increase transport and overhead costs.

One solution for supplying more seeds to farmers is through the development of more locally oriented seed enterprises. These should be able to run with less overhead and transport costs. Fiscal measures can help kick-start such businesses, e.g., import duties on seed conditioning equipment, tax holidays for the first years, etc. [60]. In addition, education and training, both on business administration and specific seed enterprise management issues, can be very important [61,62].

A special form of seed enterprise development is that of farmer cooperatives producing better seeds for themselves, for their community and then for a wider set of customers [63]. Their close connection to the farmers in their communities makes them excellent agents to respond to changing needs and good informants of public breeders. This gradually brings farmers' seed production into the realm of formal seed systems [64,65].

These forms of seed enterprise development require adapted policies [4]. Policies are commonly developed for the public or large-scale private sector seed production and distribution systems, which may not be optimally supportive to the operations of small seed enterprises and farmers' seed cooperatives [66]. Specifically, the costs of official seed quality controls can be a stumbling block for remote seed cooperatives (see below).

Such local or national seed enterprises and cooperatives commonly rely on plant varieties that were developed in the public sector.

Thus, different business models and strategies can reach different farmers with different crop seeds. High value–low volume is often aimed at, but low value–high volume (bottom of the pyramid strategies) can also provide a sustainable business model. Every seed enterprise has to provide benefits to customers who might otherwise use farm-saved seed. The benefits should translate to high quality needs from new varieties that are available at a regular interval. The seed enterprise also needs to be prepared to downgrade their seed to food grain when there is low demand in a particular year. This can only be performed when seed production costs are kept close to food grain production.

Large-scale public seed production has faced severe setbacks as has been illustrated above. There are however important roles for governments. Production of early generation seed is an important task (for the seed value chain). Good quality breeders and basic seed are necessary for bulking in subsequent growing seasons, either by commercial seed producers, or for example, for large-scale demonstrations or on-farm research work that provides access by farmers to the public varieties. There are a number of options available to create a secured supply of (pre-)basic seed for further multiplication either by the research institute itself [67] or by a special early generation seed production unit [68,69].

—Breeding requires a long-term investment and a clear vison of future needs.

For many crops, such as basic food crops and underutilized species, opportunities for private sector breeding are limited. Hence, the public sector has to invest in breeding [70]. Breeders have to work on the needs of farmers and consumers some 10 years ahead. When performed well, this makes (both public and private) breeding truly anticipatory, which contributes to resilience. This requires long-term financial commitments. Such guarantees are difficult in government accounting systems. Public sector breeders should thus have a long-term vision to make their case for continued annual budgets, even when long-term commitments cannot be secured. In private sector breeding, the commercial opportunities provide a basis for such long-term investment. Hence, plant breeder's rights (PBR) are important to secure income, which can guide investments in particular breeding programs [71,72]. However, public breeding needs caution in breeding intensity, which could focus on commercial crops and farmers. This could be detrimental to nationally important food crops that do not have financially valuable seed markets [73]. Such rights could be used to help in generating interest with seed producers to multiply a newly publicly bred variety. If they are given a (more or less) exclusive license on a new variety, they can invest in popularizing it. Seed producers cannot invest when any competitor would also be allowed to reproduce the variety that would turn out to be popular. Such exclusivity could also be operationalized through the institute's early generation seed policies when a strict certification system is operational. Where PBR can thus support both private breeding and also help public breeders to find ways to obtain their varieties to farmers through local seed companies and cooperatives, the introduction of such protection systems should thus go hand in hand with explicit research policies in line with the public task of the institutes.

Regional and international cooperation between authorities, or even harmonization, can avoid unnecessary costs [74]. Effective policies are thus essential for making commercial breeding future-proof.

—Breeding is an interdisciplinary process

Breeders, both in the public and private sector, realize that they cannot operate all by themselves. Breeding is by definition an interdisciplinary operation, as breeders need plant pathologists, agronomists and more molecular biologists on their team. In addition, breeders continually need new knowledge, which they have to obtain from outside their institution or company. Public–private and public–public collaboration with universities and international research institutes is paramount. These relationships are complex from a practical point of view. The breeder needs to be in the drivers' seat, as they have to produce something useful for the farmers. However, those with the highest academic degree and widest international experience may consider having the right to be driving. International breeding programs and international universities may be well equipped but are often far away from specific local needs. The breeder needs to be close to scientists, but even closer to the seed producers and farmers who finally determine the value of their work.

—Official quality controls are difficult to sustain

An effective phytosanitary control authority is a prerequisite for all international trade of (agricultural) products. The same capacity is needed to sustain international relations in breeding and seed markets. For variety release and seed quality control, the requirement is more specifically focused on the seed sector. The cost of full seed certification and testing can be quite high, both for the public budget and for the (emerging) seed sector itself. Variety release procedures can significantly delay access by farmers to useful germplasm.

Accreditation is one way to reduce cost, i.e., provide certain tasks to operators in the district or even to the larger seed producers who have the trained human resources under close supervision by the authorities. An advanced form of accreditation is the Quality Declared Seed (QDS) concept [75], which may provide solutions to promote seed quality in a lighter supervision regime [76,77]. In such cases, it is important to avoid undue competition between different kinds of seed suppliers. Accreditation of large seed suppliers may give them a financial advantage over smaller ones that have to pay the full cost of

official inspectors and seed testing. On the other hand, when, e.g., QDS, is applied to farmer cooperatives and full certification for small and large seed companies, this may give the cooperatives an advantage in the market.

Whatever quality management system is chosen, there is a joint interest by governments, farmers and the seed sector to fight fake seed in the market [78].

## 7. Conclusions

Farmers use a variety of seed systems for different crops, depending on their farming systems and markets. Farmers' seed systems have existed since the dawn of agriculture and have been resilient, but they increasingly face vulnerabilities. Increasing challenges as a result of fast-changing farming systems due to climate change, soil degradation and demographic trends on the one hand, and the technical and social innovations that can contribute to breeding and seed quality on the other, provide new opportunities to optimally future-proof farmers' seeds system.

Formal commercial seed systems can be quicker in their response to changes. However, they are only effective for particular crops and farmers. Formal seed systems may be slow in creating effective seed value chains beyond their original reach. Commercial seed systems are quite sustainable when appropriate policies are in place. These policies include appropriate investments in the public components such as breeding and seed research, and in many cases, aspects of quality controls. Creating a diversified, competitive seed sector in a country as well as stimulating seed entrepreneurship through fiscal and other incentives may be necessary to increase the resilience of the formal seed system.

Various interventions blur the strict distinction between formal and informal systems and create intermediate forms [3,79]: farmer groups have been formed to produce good-quality seed in more or less formal settings; others explicitly integrate formal and farmers' knowledge. The existence and complementary values of these different seed systems is not always fully recognized at the (inter)national levels. National seed policies in many countries concentrate on the formal seed systems only. Debates at the international level can sometimes give the impressions that the farmers' and community-based seed systems are the only relevant ones; or the other way around, commercial seeds should replace farmers' seeds.

Despite significant differences, the bottom line is that the resilience and vulnerabilities of both the formal and farmers seed systems are dependent on the same three common elements. The capacity of both the formal and farmers' seed systems to anticipate, absorb, respond or transform in response to shocks and opportunities are dependent on the following: First, both systems are dependent on the continuous flow and capacity to use a diversity of plant genetic resources. The flow of diversity is vital for the continuous adaptation of seeds to ever-changing demands such as for climate adaptation and food and nutrition security. Second, functioning institutions are important for the norms, policies, regulations and social relation to ensure systems of exchange and purchase, and the reliability of products and information. Trust is a vital element for both systems to function. Given the nature of seeds as an experience good, social relations and/or regulatory systems are needed to assure seed quality and reliability. Third, innovation is crucial for the identification and development of increasingly complex traits needed for plant breeding. This requires significant investment in R&D, including knowledge management.

The debate on whether formal or farmers' seed systems are best, and whether one system should replace the other, is not productive. Not a single country, nor seed system, is fully self-reliant on plant genetic resources for its food and agriculture. Given this global interdependence, it is more productive to think in terms of systems complementarity. Facilitating the continuous access to plant genetic resources, ensuring that both systems function and complement each other, and ensuring that plant breeding takes place are more decisive for the resilience of seed systems and consequently, our global food systems. Careful analysis of the opportunities and limitations of each of the seed systems is the basis

of productive decision making on how to support the many policy objectives that good seeds can contribute to.

We conclude that neither farmers' nor formal seed systems have all the answers to the challenges ahead. A variety of options are available to make these systems more resilient by themselves and to increase the chances that another seed source is available, when the other one falters. There is no one-size-fits-all in a world where nothing is static; thus, dynamic interventions need to be context specific.

**Author Contributions:** Conceptualization, N.P.L. and G.M.; methodology, N.P.L.; validation, N.P.L. and G.M.; formal analysis, N.P.L. and G.M.; resources, N.P.L. and G.M.; writing—original draft preparation, N.P.L.; writing—review and editing, N.P.L. and G.M. All authors have read and agreed to the published version of the manuscript.

**Funding:** This research received no external funding.

**Institutional Review Board Statement:** Not applicable.

**Informed Consent Statement:** Not applicable.

**Conflicts of Interest:** The authors declare no conflict of interest.

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
