# Peer review of "Seed Systems Resilience—An Overview"

_2674-1024, doi:10.3390/seeds1040028_

Round 1

Reviewer 1 Report

1) There is no affiliation.

2) The numbering of the sections is mixed up.

3) IMRAD structure is missing.

4) There is no MM section - that is, any study can be reproduced. How a systematic literature search was conducted - this should be described.

5) The necessary elements of the manuscript are missing after the references (see the template).

6) References are not designed according to a template - you need to change.

7) To increase the reader's interest, it is desirable to add summary tables and figures.

8) Statistical cluster analysis is desirable to confirm the similarity or difference between the problems under consideration.

Author Response

Dear reviewer,

We thank you for your remarks, which triggered us to significantly overhaul the paper, notably by improving the structure and re-balancing the analysis towards farmers’ an formal seed systems.

Also, we like to mention the encouragement in furthering this paper as there is a gap in the literature on the resilience aspect of seed systems that we try to fill with this review. Indeed, it is timely as various organisations, but notably FAO and CGIAR have seed systems explicitly included in their strategies. We hope that with this paper the furtherance of these strategies can be informed.

More detailed responses are below. We hope that these will satisfy you so that you can give an OK for the publication of this paper.

  • There is no affiliation.

added

  • The numbering of the sections is mixed up.

Very sorry – corrected!

  • IMRAD structure is missing.

We agree that this would be useful. We introduced a clearer research question in the introduction and a methods section 2

We consider the options for improvement – sections 5 and 6 the analysis and discussion section 

  • There is no MM section - that is, any study can be reproduced. How a systematic literature search was conducted - this should be described.

We included a more explicit MM section

  • The necessary elements of the manuscript are missing after the references (see the template).

Agree, we are sure the editor will add these

  • References are not designed according to a template - you need to change.

We’ll await detailed advice from the editor

  • To increase the reader's interest, it is desirable to add summary tables and figures.

This is not very well possible , partly because the seed systems are highly context specific, a table would run the risk of simplifying the comparison

  • Statistical cluster analysis is desirable to confirm the similarity or difference between the problems under consideration.

Since this paper does not relate to quantitative measures, we find it hard to do this; also a qualitative cluster analysis, we consider not doable/useful (see reply to question 7)

Reviewer 2 Report

The manuscript addresses a timely issue that may be adjudged as an important 'gap' in the literature. It is written in an easy-to-understand language. However, the manuscript requires some major revisions. There is no way one could assess whether the manuscript is based on any procedural and methodological rigor. I recommend that the manuscript be accepted with major revisions

Introduction

The introduction is well-written but fell short identifying the gap and questions that motivated the scoping review. As a result, it became difficult to identify and assess the contributions of the manuscript. 

Methods

Despite being a scooping review, there is still a need for a methods section in the manuscript, where the authors should discuss the research questions that guided their scoping review, as well as their information sources and literature search strategy. The elaboration should include the databases where they got the literature that informed their review, the limits that they imposed against their search strategy, and their inclusion and exclusion criterions for the publications that were considered and left out of their reviews. It is also important that the authors discuss how they did the analysis and synthesis of their data. 

Abstract

The abstract should include the gap and methods that motivated the review.

Author Response

Reviewer 2

Dear reviewer,

We thank you for your remarks, which together with thise of the other reviewers, triggered us to significantly overhaul the paper, notably by improving the structure and re-balancing the analysis towards farmers’ an formal seed systems.

Also we like to mention the encouragement in furthering this paper as there is a gap in the literature on the resilience aspect of seed systems that we try to fill with this review. Indeed, it is timely as various organisations, but notably FAO and CGIAR have seed systems explicitly included in their strategies. We hope that with this paper the furtherance of these strategies can be informed.

More detailed responses are below. We hope that these will satisfy you so that you can give an OK for the publication of this paper.

The manuscript addresses a timely issue that may be adjudged as an important 'gap' in the literature. It is written in an easy-to-understand language. However, the manuscript requires some major revisions. There is no way one could assess whether the manuscript is based on any procedural and methodological rigor. I recommend that the manuscript be accepted with major revisions

Introduction

The introduction is well-written but fell short identifying the gap and questions that motivated the scoping review. As a result, it became difficult to identify and assess the contributions of the manuscript. 

We have introduced some complementary text

Methods

Despite being a scooping review, there is still a need for a methods section in the manuscript, where the authors should discuss the research questions that guided their scoping review, as well as their information sources and literature search strategy. The elaboration should include the databases where they got the literature that informed their review, the limits that they imposed against their search strategy, and their inclusion and exclusion criterions for the publications that were considered and left out of their reviews. It is also important that the authors discuss how they did the analysis and synthesis of their data. 

Dear Reviewer, you are very right; we included a more  explicit ‘gap’  in the literature that we intend to fill (resilience of seed systems) and a methodology section 2

Abstract

The abstract should include the gap and methods that motivated the review.

included

Reviewer 3 Report

Overall, the manuscript is very well written and clear. It is easy to follow and informative. As general comments, I suggest authors to:

-          It lacks more balance on different views in terms of economic theory, free-market solution, government-private institutions roles etc. Authors often include a pure “marked solves all” idea that should be presented together with alternative points of view.

-          - Topic #4 (Options for making formal seed systems more resilient) should include solutions for the problems discussed before. However, this section mostly repeats what was presented before. Authors presented strengths and vulnerabilities well and created an expectation in relation to specific, practical solutions. It was not fulfilled as I found the topic repetitive without examples, a framework to overcome those problems. I recommend that this section to be rewritten so the manuscript reach its full potential.

 It called my attention an the lack of referencing in many parts of the text. There are several paragraphs/pages in a row that contain important statements without any citations (as an example but not restricted to L349 to 411).

There is a general lack of “real-life” examples. In the beginning some countries were cited (African ones mostly) but in the subsequent topics the manuscript become mostly theoretical. Please, include a variety of examples for each of the solutions pointed out in topic 4.

Definition of topics by numbering, bullets, underline is unclear specially in the section 3. Strengths and Vulnerabilities

Specific comments

L4 – Not sure that “good” seed is the best adjective here. It way too subjective.

L9 and 11 – futureproofing. Check consistency of the term.

L36-37 – the expression “at the right price” does not exist. It is usually defined by the supplier side in a balance in which farmers have little or no say and has many aspects build in - profit being the main one. When authors say “at the right price” it does not encompass the many difficulties that farmers face in paying for seeds, the lack of freedom in choosing seeds etc.

L16 – Is this a topic that should be numbered, highlighted?

L319-322 – A reference is required here.

L330-340 – This section seems to overlook one of the main objectives for governmental institutions to be involved in seed development that is the social role in allowing farmers to have a cheaper and free-choice alternative to expensive and sometime one-crop seeds (by “nature” or by contract enforcement).

Secondly, governmental research institutions are not for profit and, again, have a social role. We should not forget that not only farmers benefit from those institutions but also the seed industry that leverage the “cheap or free” research that are turned into profit by direct seed marketing or further development.

Thirdly, the “is not in line with economic sustainability of large-scale public seed production” should be better explained in face of the many roles that government research have.

Additionally, seed development with a smaller market (and therefore less interesting for the private sector) but important for food security is still an important role for government institutions. That should not be underestimated.  

L355 – second sentence seems to be missing brackets.

L360-362 – maybe not only number of publications, acreage or varieties but also fulfilling the gap that private companies do not fill related to less profitable seeds /smaller market share. Also, the number of families being served, the proportion of farmers less dependent on expensive seeds (debt should also be considered a criteria).

L370-373 – I agree with the interdisciplinary aspect of seed development. It also reminds me of the costs (monetary, human) that are covered by governments. As stated before, industry get “free” knowledge and seems that societies agree with that cost as part of a country’s development.

L385-387 – “challenges the economic viability of seed production of major food crops”. This is a typical biased pro-market economic statement that tries to justify public spending in private business in the name of “economic sustainability”. Is it really beyond big corporations’ capacity to help funding regulatory schemes? With profits made public, it is for the companies to justify this kind of “unsustainability”.

L96-397 – “However, this may make farmers dependent, obliterate their own seed selection and production capabilities”. That might be true but it also applicable to industry seeds. Make the tone more balanced.

L397-398 – I do not understand the meaning of the sentence in the middle of the paragraph: “Heavily subsidized seeds, by definition stands, in the way of emerging seed businesses.”

L434-442 – There is the need for more specific actions/solution. Examples?

L444-448 - There is the need for more specific actions/solution. Examples?

L449-456- There is the need for more specific actions/solution. Examples?

L455 – What are some of the possible solutions for upscaling/outscaling limitations?

L489-493 – That is a good idea but far from feasible. How would it work and how should pay for all of that? To be helpful the text could be more specific.

L520-521 – Is it a topic? The information has been mentioned before and placing it here by itself asks for more information/discussion. In short, this is cannot be a paragraph.

L534-536 – It is unclear what authors are saying here. Financing rules on public seed production is not economically sustainable? Are they for private enterprises? Is it related to taxation or lending? Please clarify and add citations.

L537-58 – “The main solution to this observed limitation is for governments to stay away from large scale operations”. Please refrain from hyperboles. Delete it and focus on your rationale. It would also be interesting adding other views on this subject despite authors’ point of view.

I wonder what authors suggest when the seed industry fails (bankruptcy, bad practices) and the seed production chain collapses. Should be public money be spent to save the industry because of their “social importance” or for being too big (important) to fail? Free market (no government) might be an economical choice but we cannot support it without considering its perils.

Author Response

Dear reviewer,

thank you for your extensive comments, which led to quite an overhaul of the paper. Detailed responses are added in the attached word file

regards Gigi and Niels

Round 2

Reviewer 2 Report

The authors have addressed my comments but I do think the manuscript should be revised to some grammar issues. 

Reviewer 3 Report

Authors corrected the manuscript sufficiently.

As suggested, the manuscript could be even more interesting if authors had included "real world" examples which would help readers to visualize the potential applications of the topics discussed.

Despite that possible improvement that was not followed, the manuscript can now be published